# Recent Advances in the Preparation and Application of Bio-Based Polyurethanes

**DOI:** 10.3390/polym16152155

**Published:** 2024-07-29

**Authors:** Ya Mo, Xiaoyue Huang, Chuanqun Hu

**Affiliations:** School of Materials and Chemical Engineering, Hubei University of Technology, Wuhan 430068, China; moya0102@163.com (Y.M.); huangxiaoyue0204@163.com (X.H.)

**Keywords:** bio-based polyurethane, biomass, preparation, functionalization, application

## Abstract

Amid environmental pollution and resource depletion, developing and utilizing biomass resources as alternatives to petroleum is a prominent research focus. Driven by environmental protection and sustainable development, the shift from petroleum-based to bio-based polyurethane is a prevailing trend in polyurethane material development. Biomass sources such as vegetable oil, polysaccharides, and lignin offer extensive application prospects in bio-based polyurethane production. Functional modifications of these polyurethanes can further expand their application range. This article explores the preparation of various bio-based polyurethanes, their applications across different fields, and their anticipated future development and uses.

## 1. Introduction

With the continuous development of science and technology, the consumption of traditional petroleum-based materials is increasing. The excessive consumption of non-renewable resources has prompted people to shift their attention to sustainable biological-based materials. Materials prepared from raw fossil matter are not sustainable and are accompanied by environmental problems. Relevant laws and policies [1] support and encourage the use of bio-based materials, which promotes their development. Bio-based materials are a type of renewable resource mainly derived from biology [2]. Their green, renewable, and environmentally friendly advantages can effectively replace traditional petroleum-based materials with high-cost-performance ones [3]. This aligns with global sustainable development strategies and has garnered significant attention from researchers.

Polyurethane was first invented by German chemist Otto Bayer in 1937 and began commercial production in the 1940s. With progress in research and technology, the application of polyurethane materials gradually expanded. It is a polymer material with a microphase-separated structure [4] generated by the reactions of isocyanates with polyols, polyamines, and other polyhydroxy compounds (as shown in Figure 1), which gives these materials excellent characteristics, such as good mechanical properties [5], biocompatibility [6], corrosion resistance [7], and many others. This allows them to be widely used in flexible electronics, adhesives, anti-corrosion materials, and other fields [8,9,10]. The traditional raw materials of polyols and isocyanates mainly come from petroleum resources. With increasing concern about their environmental impact, the traditional production process and waste disposal of polyurethane materials have gradually become focuses. In order to reduce dependence on limited resources and reduce environmental impact, people have begun to focus on the development of renewable resources, such as bio-based ones, to produce economically and environmentally friendly polyurethane materials [11]. At present, polyurethane has undergone a comprehensive transformation from petroleum-based to bio-based. Renewable biomass resources such as vegetable oil [12], starch [13], polysaccharides [14], lignins [15], and agricultural waste [16] are used to replace the petroleum-based raw materials in it.

The research of bio-based polyurethane is currently in a stage of active development, and researchers are exploring various renewable biomass resources as raw materials for its preparation, optimizing the synthesis route and process conditions so as to improve the preparation efficiency and quality stability of bio-based polyurethane and explore its practical value in various industrial and consumer applications. In this paper, the preparation of different biomass material-based polyurethanes and their applications in different fields are reviewed. Finally, this paper looks forward to the development of new fully bio-based polyurethane materials, aiming to contribute to the research of these materials and broaden their application scope.

## 2. Preparation of Bio-Based Polyurethane

Bio-based polyurethane (BPU) is a kind of polymer material made from renewable resources that has received extensive attention due to its environmental friendliness and renewability. It not only reduces dependence on petroleum resources but also has a certain biodegradability [17], which is of great significance in environmental protection.

### 2.1. Vegetable Oil-Based Polyurethane

At present, research on vegetable oil-based polyurethanes is gradually increasing, focusing on the development of sustainable and environmentally friendly alternative materials. These oils are often renewable resources and have lower carbon footprints than conventional petroleum-based polyurethanes. Vegetable oil has become the main raw material that replaces petroleum-based materials in synthesizing polyols and isocyanates [18,19]. Many vegetable oils, such as castor oil, soybean oil, tung oil, and palm oil [20,21,22,23], are prepared as vegetable oil-based polyols through epoxidation and hydroxylation [24,25] and further synthesized into polyurethane materials.

#### 2.1.1. Castor Oil

Castor oil has high biodegradability properties [26]. Its molecular structure contains hydroxyl groups, making it a good choice for preparing bio-based polyurethane materials. It can form a cross-linked structure with the polyurethane molecular chain to improve the performance of polyurethane materials. Lee et al. [27] prepared castor oil (CO)-based polyols with high functionality through a thiol-ene reaction (as shown in Figure 2) and used the co-based polyol mixture to prepare polyurethane foams through a foaming method. The mechanical properties of the foam were significantly enhanced, with a compressive strength of up to 571 KPa, and the thermal stability was also significantly improved. Su et al. [28] used castor oil to synthesize 1-thioglycerol-modified castor oil-based polyols with a high hydroxyl value of 463 mg KOH g^−1^ through a solvent-free thiol-olefin reaction and prepared a cross-linked polyurethane. The tensile strength of this polyurethane could reach 83 MPa, its Tg temperature was 124 °C, and it had good optical properties. Tuo et al. [29] converted castor oil into castor oil-based triacrylate structures (MACOGs) through chemical modification and then prepared castor oil-based aqueous polyurethane acrylate emulsions. Through the preparation of aqueous polyurethane coatings through UV curing, with an increase in MACOG content, the glass transition temperature of the sample increased from 20.3 °C to 46.6 °C and the surface water contact angle increased to 90.57°.

#### 2.1.2. Soybean Oil

Soybean oil is mainly composed of triglycerides, which contain unsaturated fatty acids such as linoleic and linolenic acid. These acids give soybean oil wide application potential in the polyurethane field. Lou et al. [30] used epoxidized soybean oil to react with 1,3-propanediol, trimethylolpropane, and xylitol, synthesizing a series of soybean oil-based polyols and preparing waterborne polyurethane. That study investigated the effect of the hydroxyl value of soybean oil-based polyols on polyurethane performance. The test results showed that increasing the hydroxyl value content improved the Tg temperature (increased from 30.04 °C to 88.28 °C) and water contact angle (increased to 98.20°) of the polyurethane material while reducing its water absorption. Li et al. [31] used chlorinated soybean oil (SOL) as a raw material to synthesize a novel soybean oil-based polyurethane acrylate polyol (ADSO) and prepared soybean oil-based waterborne polyurethane (ADSO-WPU). Through the use of graft copolymerization with methyl methacrylate (MMA), butyl acrylate (BA), and 2-(perfluorohexyl) ethyl acrylate (PFOMA), grafted fluorinated acrylate soybean oil-based waterborne polyurethane (ADSO-PUA) was also obtained. ADSO-PUA had higher hardness (6H) and mechanical properties (9.79 MPa) and better thermal stability (T5/°C increased to 193 °C) than ADSO-WPU. Dai et al. [32] used epoxy soybean oil and castor oil acid to prepare soybean oil-based polyols (as shown in Figure 3). The resulting polyols were mixed with isophorone diisocyanate, dimethylolpropionic acid, trimethylamine, and ethylenediamine to synthesize polyurethane emulsions with different R values (-NCO/-OH molar ratio). As the R value increased, the tensile strength, thermal stability, glass transition temperature, and hydrophobic angle of the polyurethane improved.

#### 2.1.3. Tung Oil

Tung oil, a vegetable oil extracted from the seeds of the tung tree, is mainly composed of triglycerides. The primary characteristic of tung oil is its high conjugated linoleic acid (eleostearic acid) content, which is the source of its unique properties. Li et al. [33], using tung oil and 4-maleimide phenol as raw materials, synthesized a tung oil-based polyphenol (ATOM) containing phenolic hydroxyl groups through the Diels–Alder addition reaction and prepared a thermosetting polyurethane with a polyurethane prepolymer. This polyurethane had a good self-repairing function, and its elongation at break could be restored to more than 80% after self-healing of the fracture. Zhao et al. [34] prepared a tung oil-based polyphenol and used it as a raw material to prepare a polyurethane foam and film through thermal compression molding. The film had adjustable mechanical properties and could be recycled, which is beneficial to realizing the green production of polyurethane materials (as shown in Figure 4). Zhou et al. [35] prepared a phosphorus-containing tung oil-based polyol (PTOP) and a silicon-containing tung oil-based polyol (PTOSi) through ring-opening reactions and prepared two kinds of polyurethane foams (RPUFs), respectively. RPUFs have good flame retardancy and thermal stability. When the content of PTOP or PTOSi was 80%, the limiting oxygen index of an RPUF could reach 24.0% or 23.4%, respectively, and its compressive strength could reach 0.82 MPa or 0.25 MPa, respectively, given an appropriate content. Its thermal conductivity increased with increases in the PTOP and PTOSi contents. He et al. [36] used a simple one-step purification process to prepare a tung oil-based polyol (TOP), which was used to prepare an aqueous polyurethane dispersion (WPUD). The WPUD had good mechanical properties and hydrophobic ability, with a tensile strength of 16.8 MPa and a water contact angle of 109.5°, and could achieve self-assembly of colloidal particles under certain conditions.

### 2.2. Polysaccharide-Based Polyurethane

Polysaccharide-based polyurethane is an important kind of bio-based polyurethane material for which research is developing and expanding constantly. Its main raw materials come from various polysaccharides, such as starch, cellulose, and chitosan. These polysaccharides usually come from renewable sources. In general, polysaccharide-based polyurethane, as a new bio-based polyurethane material, is attracting more and more attention and research. With advancements in technology and the increasing demand for sustainable development, it is expected that it will have wider applications and market prospects in the future.

#### 2.2.1. Starch

Starch is a natural polysaccharide composed of glucose units linked by α-1,4-glycosidic bonds, which can be divided into amylose [37] and amylopectin [38]. It is widely present in plants and has abundant resources and renewability. Converting starch to polyurethane materials can provide an environmentally friendly and sustainable option. Zhou et al. [39] used aminosulfonic acid as a catalyst to liquefy starch and prepare starch-based polyurethane coatings. That study of polyurethane-curing reaction kinetics showed that the starting, peak, and ending temperatures of the curing reaction were 23.31 °C, 83.15 °C, and 199.75 °C, respectively. The activation energy of the reaction system was 44.34 kJ mol^−1^, the pre-exponential factor was 7.36 × 103 s^−1^, and the reaction order was 0.88. This provides a theoretical basis for the actual curing process of starch-based polyurethane materials. Liang et al. [40] used bio-based acetylated starch (ac starch) as a polyol and a hexamethylene diisocyanate trimer (HDIT) as a cross-linking agent to prepare a novel fully hydrophobic polyurethane (PU) coating. A small amount of monomethanol-terminated polydimethylsiloxane was introduced as a lubricating component. The coating had high transparency (>97%), excellent anti-fouling properties, and could provide corrosion resistance for metal surfaces (as shown in Figure 5). Lubcazk et al. [41] reacted starch with allyl carbonate to produce polyether alcohol and prepare a rigid polyurethane foam. The apparent density, water absorption rate, and polymerization shrinkage of the polyurethane foam were similar to those of classical polyurethane foams, and it had high thermal resistance, able to be heated for a long time at 175 °C. It has good application prospects in the field of insulation materials. Lubcazk et al. [42] also used starch as a raw material and formaldehyde, glycerol, and allyl carbonate as functionalizing agents to synthesize oligomers, which were used to prepare a polyurethane foam on this basis. This kind of polyurethane foam had good heat resistance, its mechanical properties were stronger after thermal exposure, and it could be heated for a long time at 200 °C.

#### 2.2.2. Cellulose

Cellulose is a linear polysaccharide formed by the connection of glucose through β-1,4-glycosidic bonds. Each glucose unit is connected by covalent bonds to form long chains. Cellulose has a supramolecular structure, and its solid state is represented by crystalline and amorphous regions [43]. It can be used as a bio-based raw material for the preparation of biodegradable and environmentally friendly polyurethane matter. Maiuolo et al. [44] used cellulose-derived polyols as chain extenders and citric acid cellulose as a thickener to prepare a series of new bio-based polyurethane composite foam materials. The hyperbranched structure of citric acid cellulose introduced into the polyurethane chain increased the hydrogen bonds in the PU system, resulting in a compressed volume of the composite material and a more compact structure. The bio-based polyurethane foam and its prepared composites had good mechanical properties. Szpilyk et al. [45] synthesized polyols in water, using cellulose, triglycerides, and ethylene carbonate as raw materials, and prepared a rigid polyurethane foam. This foam had obviously excellent density, water absorption, and polymerization shrinkage rates, and heat resistance. Since the prepared polyols were biodegradable, the resulting polyurethane foam had the property of biodegradability. Szpilyk et al. [46] also used cellulose hydrolysis products as raw materials, catalyzed by potassium carbonate, to react with triglycerides and ethylene carbonate and produce polyols. These polyols were used for the preparation of a polyurethane foam. The polyurethane foam had good heat resistance, and its mechanical properties were enhanced under long-term heating at 175 °C, with double the compressive strength compared with the initial foam. Hou et al. [47] used amorphous regenerated cellulose pulp (RCP) as a raw material and adopted a green method of hydroxyl/isocyanate chemistry to prepare thermoplastic cellulose-grafted polyurethane (RCP-g-PU). The resulting polyurethane could be directly thermally pressed into a flexible and foldable transparent film (as shown in Figure 6).

#### 2.2.3. Chitosan

Chitosan is a natural polysaccharide derived from the deacetylation of chitin [48,49]. It is composed of N-acetylglucosamine and glucosamine units linked by β-1,4-glycosidic bonds. Due to its rich hydroxyl and amino groups, it can be used to prepare bio-based polyols. Gao et al. [50] prepared chitosan-based polyurethane (c-PU) microencapsulated phase-change materials (MicroPCMs) through the interfacial polymerization of hexamethylene diisocyanate and chitosan, assisted by charge attraction (as shown in Figure 7). This material exhibited excellent latent heat performance (ΔH-m = 106.3 J/g, ΔH-c = 105.1 J/g), high energy storage efficiency (E = 71.4%), excellent thermal stability, and cyclic durability, as well as reversible photochromic ability. Strzalka et al. [51] hydroxylated chitosan with glycerol and ethylene carbonate under different environments to obtain a polyurethane foam using hydroxylated chitosan. This polyurethane foam had a similar performance as a typical rigid PUF, with increased thermal resistance, enhanced compressive strength after heat treatment at 150 °C, and good thermal stability at 175°C for a long time. Javaid et al. [52] used an -NCO end-terminated prepolymer and chain extender (1,4-butanediol/starch/chitosan) to synthesize starch/chitosan-modified polyurethane (PU). When the mixed amount of starch and chitosan was equal, the thermal stability of the PU was higher. The combination of starch and chitosan provided an efficient pathway for preparation of the PU.

### 2.3. Lignin-Based Polyurethane

Lignin-based polyurethane is synthesized using lignin as the main raw material. Lignins are natural polymers present in wood fibers, and their renewability and richness make them a green and sustainable polyurethane raw material choice. As a potential bio-based material, lignin-based polyurethane is attracting more and more attention in research and industrial applications.

Lignins are complex organic polymer compounds that, together with cellulose and hemicellulose, constitute the framework of the plant cell wall. The structure of lignins is very complex, and they are the second most abundant source of renewable carbon with aromatic structures [53]. Their reactivity can be improved through chemical modification, such as graft polymerization and epoxidation [54,55,56], allowing them to partially replace petroleum-based materials [57]. Li et al. [58] used lignin macromolecules as polyhydroxy structural materials to synthesize polyurethane macromolecules. The tensile strength and elongation at break of this polyurethane reached 16.2 MPa and 1049.5%, respectively. Lignin acted as a fulcrum to form hydrogen bonds with the polyurethane macromolecular chains, creating a hydrogen-bonded network structure that allowed for a 94.7% elastic recovery rate while maintaining good mechanical properties after thermal reprocessing. Zhang et al. [59] reported a new method of ozonizing lignin to obtain an oxidized lignin (OL), which was then used to partially replace PEG200 in reacting with isocyanate to produce a lignin-based PU prepolymer. After curing, an oxidized lignin-based PU (OLPU) was obtained. Due to the rigid aromatic structure and cross-linking of lignin, the OLPU exhibited excellent properties, with a tensile strength of 47.2 MPa, hardness of 2H, good hydrophobic performance, and thermal stability. Wang et al. [60] used the reaction between a phenolic aldehyde lignin (PL) and isocyanate as the basis of the construction of a strong and reprocessable biopolyurethane elastomer (PLPU). Dynamic hydrogen bonding and phenolic carbamates formed an adaptive cross-linked network, giving the PLPU adjustable mechanical properties, a tensile strength of 58.8 MPa, a modulus of 1350.3 MPa, a toughness of 57 MJ/m^3^, and good elastic recovery capabilities. Vieira et al. [61] used eucalyptus lignin to obtain lignin-based polyols (LBPs) through an oxygen alkylation reaction with propylene carbonate, and then used these to prepare polyurethane adhesives. Compared to commercially available polyurethane adhesives (CPAs), the lignin-based PU adhesives had better chemical resistance and adhesive efficiency and a shorter gel time, making them highly promising for use in wood adhesive applications. Huang et al. [62] introduced weak hydrogen bonding, strong hydrogen bonding, and dynamic siloxether covalent bonding to prepare a high-strength, heat-resistant, and recyclable lignin-based polyurethane (LPU) (as shown in Figure 8). The covalent bonds between lignin and the matrix promoted their compatibility, forming a strong cross-linked network. Through multiple layers of dynamic interactions, the mechanical strength of the prepared elastomer reached 73 MPa, with an initial thermal decomposition temperature of 310.5 °C and good recyclability. Under the conjugated structure provided by lignin, the elastomer had good light-to-heat conversion capabilities.

## 3. Application of Bio-Based Polyurethanes

Bio-based polyurethane materials mainly come from renewable biomass materials. Compared to traditional petroleum-based polyurethanes, bio-based polyurethanes are sustainable, with a certain biodegradability, and resistant to chemicals [63,64,65,66], which is all beneficial for reducing environmental pollution. By modifying bio-based polyurethanes, specific properties can be given to them, making them widely applicable in flame-retardant materials, food packaging, self-repairing materials, sensors, and other fields [67,68,69,70].

### 3.1. Flame-Retardant Materials

Bio-based polyurethane flame-retardant materials are made by introducing, for example, halogen-free, phosphorus-based, or nitrogen-based flame retardants into bio-based polyurethane. Zemla et al. [71] used rapeseed oil as a raw material to prepare a rigid polyurethane foam (RPURF) containing bio-based polyols and different phosphorus-based flame retardants (triethyl phosphate (TEP), dimethyl phosphate propanoate (DMPP), and cyclic phosphoric acid esters). All the obtained foam materials had low thermal conductivity coefficients and limit oxygen indexes (LOIs) of above 21 vol%. The modified RPURF had a lower fire development tendency and could be classified as a self-extinguishing material. Zhang et al. [72] used citrus peel oil-based derivative limonene dithiol (LDM) and glycerol-1-allyl ether (GAE) as raw materials and synthesized bio-based polyols (LDM-GAE) through a one-step photochemical thiol-ene reaction. The synthesized polyols were used to prepare flame-retardant polyurethane foams with dimethyl phosphate (DMMP) as an additive flame retardant (AFR) and a novel bromine-containing reactive flame retardant (RFR) derived from 2,4,6-tribromophenol. These two flame-retardant polyurethane foams had excellent mechanical properties, with short self-extinguishing times (7.5 s and 12.8 s, respectively) and low mass loss after combustion. Wang et al. [73] used a phosphorus–nitrogen chain extender [(bis(2-hydroxyethyl)amino]-methylphosphonic acid dimethyl ester (BH)] to modify a novel bio-based WPU dispersion prepared from castor oil and soybean polyols. The results of their performance characterization showed that the WPU had an LOI of up to 28.1%, a tensile strength of up to 8 MPa, and a Young modulus of up to 62.3 MPa, indicating good mechanical properties. Akdogan et al. [74] liquefied beet pulp as a source of bio-based polyols (SBpol) for the production of a bio-based rigid polyurethane foam (sPUF) (as shown in Figure 9). Flame-retardant materials were prepared by adding expandable graphite (EG) and/or dimethyl phosphoric acid methyl ester (DMMP) to an sPUF. EG, acting in the condensed phase, and DMMP, mainly acting in the gas phase, could improve the flame-retardant properties of the sPUF, with an LOI of up to 24.9%. Fidan et al. [75] prepared bio-based rigid polyurethane foam composites (RPUFc) by liquefying apricot kernel shells. This RPUFc series had an LOI of up to 18.5% while also having good mechanical properties, with a compressive modulus of 180.3 KPa and a compressive strength of 14.9 KPa. This flame-retardant material can be used in the fields of construction and insulation materials. Akdogan et al. [76] used a novel active-phosphorus sunflower oil-based bio-based polyol to prepare bio-PUFs and added expandable graphite (EG) and dimethyl phosphoric acid methyl ester (DMMP) to them to improve their flame-retardant properties. The thermal conductivity coefficients of the bio-PUF composites were in the range of 28.72 to 32.51 mW/(mK), and the LOI value could reach 25.7%, which was about 32% higher than that of a compared petroleum-based PUF. Gong et al. [77] prepared a reactive hyperbranched flame-retardant polyol (DOPO-masi) containing three flame-retardant elements, P, N, and Si, which was mixed with expandable graphite (EG) to obtain a plant oil-based flame-retardant polyurethane foam (RPUF) with multiple flame-retardant systems. This foam had excellent flame-retardant and smoke-suppression properties, with an LOI of 30%, and good mechanical properties. Bio-based polyurethane flame-retardant materials have great commercial value in fields such as building materials and fireproof coatings [78,79].

### 3.2. Food Packaging

In order to alleviate the environmental burden caused by the difficulty of degrading traditional plastics, the transition from traditional packaging materials to biodegradable ones has arisen. Biodegradable food packaging is gradually replacing its traditional plastic counterparts [80,81]. Bio-based polyurethane materials have biodegradable properties, which help to reduce the impact of packaging waste on the environment. Chen et al. [82] synthesized a bio-based waterborne polyurethane (PLA-WPU) with double salt. The hydrophobic modified cellulose nanocrystal (M-CNC) and polycarbonate diamine (PDCDA) synergistically enhanced the PLA-WPU, which had a water contact angle that could reach 143.5° and excellent waterproof performance. It could maintain these, as well as excellent mechanical properties and high barrier performance, under extreme conditions. After 8 days of degradation in a lipase PBS solution, the degradation rate reached 54%. Fan et al. [83] used soybean oil-based polyols to prepare polyurethane foams by adding corn protein to improve the ethylene adsorption rate. The foams had 3D porous structures with good hydrophobicity and biodegradability and efficient ethylene adsorption. This innovative packaging material provides advanced ethylene adsorption and cushioning capabilities, which in turn provide an effective and novel method for the rational design and manufacture of advanced bio-based fruit packaging. Zheng et al. [84] used a simple and environmentally friendly electrostatic self-assembly strategy to prepare a series of castor oil-based waterborne polyurethane/polyhexamethylene guanidine (WPU/PHMG) composites (as shown in Figure 10). These composite materials had antibacterial rates of over 99.9% for both Staphylococcus aureus (*S. aureus*) and Escherichia coli (*E. coli*), as well as stable and long-lasting antibacterial properties, low water absorption rates, and good mechanical properties, giving them good application prospects in the field of food packaging. Arilk et al. [85] reported the preparation of bio-based thermoplastic polyurethane (TPU) fiber scaffolds containing essential oils (EOs). Studies have shown that the addition of essential oils increases the fiber diameter and reduces surface roughness, resulting in a lower contact angle of a composite fiber. This shows the advantages of essential oil incorporation into electrospun fibers in terms of morphology and size range, which can be applied to the field of food packaging. Indumathi et al. [86] used sesame oil-based polyurethane (PU) and chitosan (CS) as raw materials and added zinc oxide nanoparticles in different proportions to prepare biodegradable food packaging films. The addition of nano-ZnO enhanced the antibacterial, barrier, and hydrophobic properties of the film. After the film was placed in soil for 28 days, the weight loss rate was 86%. Compared to commercial polyethylene films, this film effectively reduced bacterial contamination.

### 3.3. Self-Healing Materials

The research on green and self-repairing polyurethane materials meets the demand for sustainable development. The self-repairing effect of polyurethane mainly comes from its reversible covalent bonds, disulfide bonds, ester exchange reactions, etc. [87,88,89,90]. Bio-based polyurethane molecules often contain multifunctional groups that can react with other molecules or form dynamic bonds to promote the self-repair process. They also have a cross-linked network structure that provides stability and strength to the material. Lee et al. [91] synthesized a eugenol dimer (EGD) and mixed it with a mixture of polyols (tetramethylene ether glycol) and 4,4′-methylenediphenyl diisocyanate prepolymer to synthesize a PU. This EGD-PU exhibited a self-healing ability of up to 99.84%, with an 87.71% self-healing rate after three cycles, and a good antioxidant capacity. Zhang et al. [92] used 1,8-methanediamine (MTDA) and CO2 catalytic carbonation to synthesize a hyperbranched bio-based cyclic carbonate (Ec-MTDA) as a raw material to prepare a potent, self-healing, and catalyst-free NIPU (ECMP). This ECMP had good tensile and adhesive capabilities, and due to the dynamic transesterification reaction between the aminocarbamate and hydroxyl groups, it also had good self-healing, reprocessing, and shape memory performance, with a self-healing rate of up to 91%. Xu et al. [63] synthesized a self-repairing/renewable polyurethane based on bio-based vanillin and tyrosine. The use of two different thermally reversible mechanisms (deblocking/reblocking of phenolic urethane and imine metathesis) allowed the structure of the material to change under specific temperature conditions, resulting in versatility and a self-healing ability in applications, with strong controllability and reversibility. The resulting PU had considerable self-healing efficiency and renewability, with its tensile strength recovering to nearly 95% within 2 h. After it was remolded five times, its chemical structure, glass transition temperature, deblocking behavior, tensile strength, elongation at break, and gel content remained unchanged. Du et al. [93] used lignin as a polyurethane raw material, combined with the Diels–Alder (DA) reaction and hydrogen bonds, to prepare a self-healing lignin-based polyurethane (PUDA-L). Due to the internal DA and cross-linked hydrogen bonds, the PUDA-L had good thermal stability, good fatigue resistance, a good shape memory effect, excellent mechanical strength, and excellent self-healing ability. Its self-healing rate reached 100% after being melted and heated at 130 °C for 4 h. The unesterified hydroxyl group formed cross-linked hydrogen bonds with the carbamate and DA structures. The hydrogen and DA bonds significantly improved the mechanical and self-healing properties of the elastomer, as the healing rate reached 101.38% and the maximum stress after healing reached 29.46 MPa (as shown in Figure 11). Ryu et al. [94] used polybutylene furanate and bismaleimide as raw materials to synthesize a furan-based self-repairing polymer through the Diels–Alder reaction and mixed it with biopolyurethane to prepare a self-healing polymer film. This film achieved self-healing within 24 h, with a self-healing rate of up to 90%, and was biodegradable. Liu et al. [95] proposed a method for preparing thermally induced dynamic phenolic aminocarbamate cross-linked thermosetting polyurethane, noting that tannic acid (TA) formed a phenolic aminocarbamate network with the isocyanate groups. The prepared TA-based polyurethane (TA-PU) had good mechanical properties and self-repairing, reprocessing, and shape memory functions. Through adjustment and control of the dynamic phenol aminocarbamate bond content in the polyurethane backbone, a high self-healing efficiency was achieved, with a self-healing rate of over 90%. DA covalent bonds have the advantages of high selectivity and mild reaction conditions, so they are widely used. However, compared to other reversible bonds, their reversibility is poor and the reaction conditions are harsh.

### 3.4. Sensing Materials

Bio-based polyurethanes exhibit multi-directional advantages in the field of sensing materials, making them ideal choices for developing the new generation of these materials. These polyurethanes can achieve sensing functions by introducing specific chemical molecules or functional groups. These groups can be chemical sensors that respond to specific substances or conditions in the environment. Bio-based polyurethanes can also be designed to be bioresponsive sensors: biosensors that respond to changes in the environment within a particular biomolecule or organism. They can be used in food safety testing, medical health, smart materials, and so on. Yusoff et al. [96] developed a new bio-based ion-selective, nitrate-sensing polycationic polyurethane membrane. Electrochemical impedance spectroscopy (EIS) testing showed that after ion exchange of iodide and nitrate on polycationic polyurethane, its conductivity increased from 2.84 × 10^−11^ to 5.34 × 10^−11^ S cm^−1^, with a sensitivity of up to 5.94 × 10^−2^ μW/ppm. It was suitable for nitrate detection in bio-based sensing materials due to its fast detection speed and good repeatability. Khalifa et al. [97] used a solution casting method to prepare flexible graphene and bio-based thermoplastic polyurethane (TPU) films and studied their piezoresistive sensing properties. With increasing graphene nanocontent, the thermal stability, tensile properties, and electrical conductivity of the material were significantly improved, with a sensing coefficient of 11, and it still exhibited high stability and repeatability after >10,000 bending cycles. Deng et al. [98] introduced bundled cellulose nanocrystals (TCNCs) into castor oil-based aqueous polyurethane (WPU) to prepare bio-based nanocomposites. Due to the excellent binding ability of TCNCs with high aspect ratios, they were observed through TEM to bridge adjacent WPU microspheres together, forming a dispersion fiber three-dimensional network. The random, uniform distribution of the TCNCs in the WPU could be observed by SEM, and the distribution of TCNCs with high content was denser on the surface of WPU films. This also guaranteed the excellent properties of the composite (as shown in Figure 12). This material had excellent thermophysical (storage modulus: 62.06–903.32 MPa) and mechanical properties (tensile strength: 7.36–18.29 MPa; elongation at break: 110–315%). Through coating with silver nanowires, a flexible strain sensor with fast responses, high sensitivity, and good cycling stability was created, suitable for human motion monitoring. Li et al. [99] used carbonized soybean oil (CSBO) and polyurethane (PAPMS) as raw materials to prepare conductive bio-based non-isocyanate polyurethane (NIPU). The mechanical properties and electrical conductivity of the NIPU were adjusted by changing the molar ratio of the amino groups to that of cyclic carbonate and adding carbon black. This material had excellent thermal stability, solvent resistance, and tensile properties, an excellent self-healing ability, and exhibited good sensing performance and stability over a wide temperature range of −40 to 60 °C. Liu et al. [100] synthesized silicone-sealed polyurethane (Si-BPU) with high stretchability and degradability and combined it with carbon nanotubes (CNTs) to create a fiber strain sensor with a strain range of 0–353.3%, a gauge factor (GF) of 4513.2 at 353.3% strain, and reliable stability over 10,000 repeated stretching–releasing cycles, with a response time of < 163 ms. Zhang et al. [101] synthesized a hydrogen-bonded polyurethane urea elastomer (PUSS) using o-aromatic diaminodisulfide and 1,3-dihydroxyacetone extracted from biomass as key components. They combined MXene and carbon nanofibers into a PUSS matrix to produce a resistive sensor with strong mechanical properties, high sensitivity, and excellent self-healing ability.

Bio-based polyurethane sensors may encounter some technical and application obstacles, such as stability, durability, sensitivity, selectivity, biocompatibility, and safety, when integrated into actual devices. Overcoming these obstacles requires a combination of materials science, sensing technology, engineering and manufacturing, and application requirements. With advancements in technology and the increasing demand for sustainability and intelligence, bio-based polyurethane sensors are gradually overcoming these challenges and will be more widely used in the future.

### 3.5. Other Applications

Biopolyurethane has a wide range of application prospects in other fields, such as in biomedicine, agriculture, adhesives, and electromagnetic shielding, due to its good biocompatibility and excellent performance. Bio-based polyurethanes also have good application prospects in tissue engineering and regenerative medicine and can be used as materials to manufacture artificial tissues and organ scaffolds, support cell growth and tissue regeneration, and promote the development of tissue repair and regenerative medicine. Yadav et al. [102] used alginate (SA), bio-based polyurethane (BioPU), and other materials to prepare a scaffold for wound care. A sponge surface provided a suitable wet adhesive physical environment to support the adhesion and growth of skin cells (HaCaT cells) and showed non-toxic and in vitro biocompatibility. It has great application potential in tissue engineering. Bio-based polyurethane can be used as a coating agent in the manufacturing of controlled-release fertilizer. This fertilizer can control the release rate and time of the nutrients through the polyurethane coating so as to improve the utilization efficiency of the nutrients and reduce the loss of nutrients and environmental pollution. Yu et al. [103] used soy protein isolate (SPI) and polylactic acid–hydroxyacetic acid as raw materials, prepared nano-scale soy protein microcapsules (SMCs) through water-in-oil-in-water emulsification technology, and prepared a nano-scale soy protein microcapsule self-repairing biopolyurethane-encapsulated controlled-release fertilizer (NSBCF) with enhanced controlled release performance. Compared to the unmodified BPCF, the controlled release life of the NSBCF was extended by more than 28 days through its modified self-repairing biopolymer coating. After the release of SPI from the SMCs, the coating could self-repair by reacting with pentanal to form solid resin and block the micropores. The self-healing process of the SMCs successfully reduced the nutrient release rate and extended the nutrient release life of the NSBCF. Because of their sustainability, high performance, and wide application potential, bio-based polyurethane adhesives have obvious competitive advantages and development prospects in the adhesive market. Du et al. [104] used bio-based vanillin oxime (VO), soybean oil polyol (SBOH), polytetrahydrofuran (PTMG), and isophorone diisocyanate (IPDI) as raw materials to construct a bio-based dynamic cross-linked polyurethane hot melt adhesive (DPU). After 30 min of curing, the maximum shear strength of the DPU reached 6.55 ± 0.88 MPa, and it had good self-healing and reusable adhesive properties. After seven complete cycles of breaking and repairing, the self-healing rate could reach 80%. Bio-based polyurethanes have shown broad application prospects in the field of electromagnetic shielding, and their advantages in forming ability, mechanical properties, electrical properties, and environmental friendliness make them some of the most important electromagnetic shielding materials for the future of the electronics, communications, medical, and military fields. Selvaraj et al. [105] used castor oil-based polyurethane, graphite nanosheets (GNPs), zirconia (ZrO_2_), and bamboo charcoal (BC) to manufacture a bio-based, low-density EMI defensive material. The maximum EMI SE of this material was 28.03 dB in the 8–12 GHz frequency band.

In summary, bio-based polyurethanes have a wide range of application prospects in many fields due to their greenness, sustainability, and many other advantages.

## 4. Conclusions and Outlook

As a kind of sustainable and environmentally friendly high-performance material, bio-based polyurethanes have extensive research progress and application prospects, providing an important choice and development direction for replacing traditional petroleum-based polyurethane. Many companies and research institutions are actively promoting the commercialization process of bio-based polyurethane, although there are still challenges in mass production and cost-effectiveness, but with technological advances and market demand growth, it is expected that more bio-based polyurethane products will be put on the market in the future. This article summarizes the research progress for plant oils, polysaccharides, and lignins in the preparation of biopolyurethane, introduces the current application statuses of bio-based polyurethanes across various fields, and explores the development and application prospects thereof.

Currently, the development of bio-based polyurethanes faces several challenges. Firstly, for example, the high costs of bio-based raw materials will result in relatively high production costs for bio-based polyurethanes, limiting their large-scale commercial applications. Secondly, although bio-based polyurethanes perform well in many aspects, there are still gaps in some properties compared to traditional petroleum-based polyurethanes. Thirdly, achieving high performance while ensuring environmental protection and sustainability remains a challenge.

Bio-based polyurethanes have good biocompatibility. Through molecular design and modification technology, they can be functionalized to have enhanced biocompatibility and biodegradability and improved recyclability. This can increase their applicability in the fields of biomedicine and human detection, providing a basis for continuous research.

## Figures and Tables

**Figure 1 polymers-16-02155-f001:**
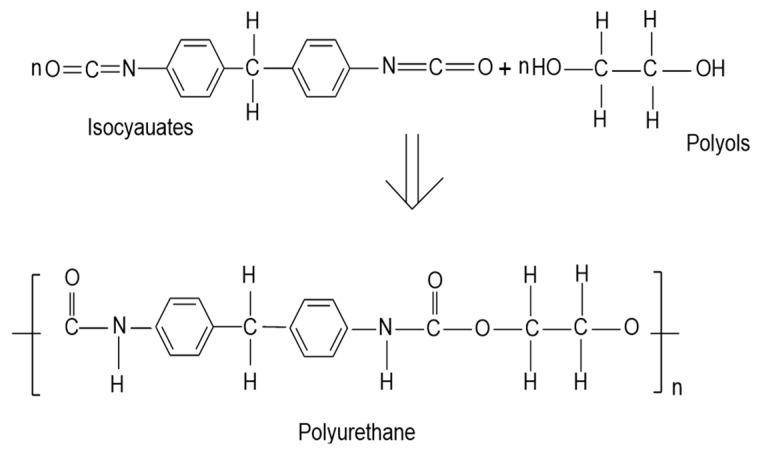
Polyurethane synthesis.

**Figure 2 polymers-16-02155-f002:**
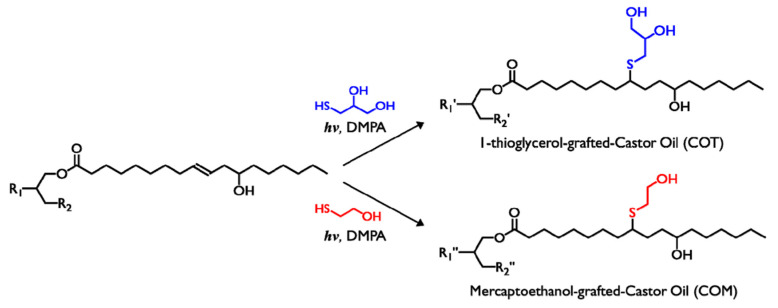
Preparation of castor oil-based multifunctional polyols using the thiol-ene photo-click reaction [27].

**Figure 3 polymers-16-02155-f003:**
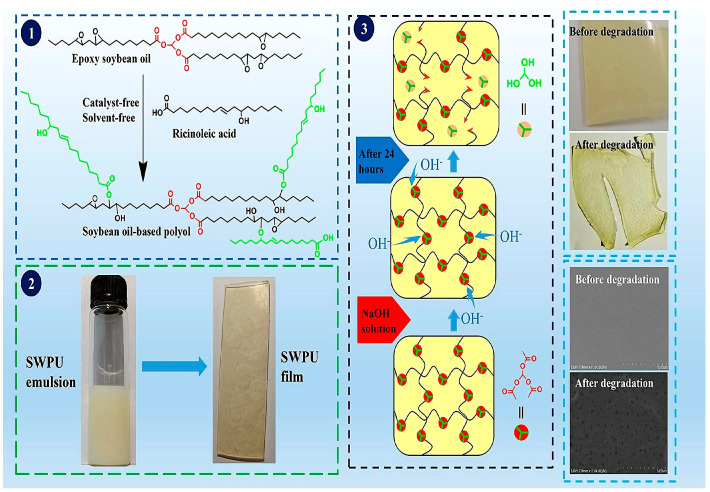
Preparation and degradation performance test of SWPU (1: Preparation of soybean oil-based polyols; 2: Preparation of SMPU; 3: Biodegradability of SWPU) [32].

**Figure 4 polymers-16-02155-f004:**
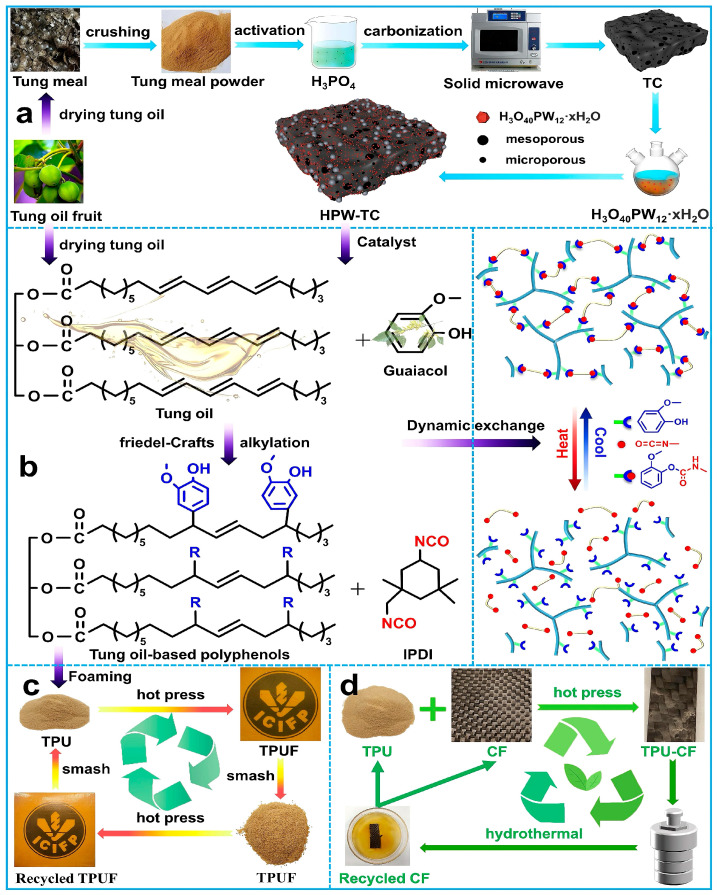
Schematic illustration of the preparation, utilization, and sustainable recycling of TPU foams. (**a**) Schematic illustration of preparing an HPW-TC catalyst from tung oil fruit after extracting the tung oil. (**b**) Schematic illustration of preparing tung oil-based polyphenols with HPW-TC as the catalyst. (**c**) Schematic illustration of a sustainable recycling concept for TPU foams. (**d**) Schematic illustration of the preparation and recyclability of TPU-CF composites [34].

**Figure 5 polymers-16-02155-f005:**
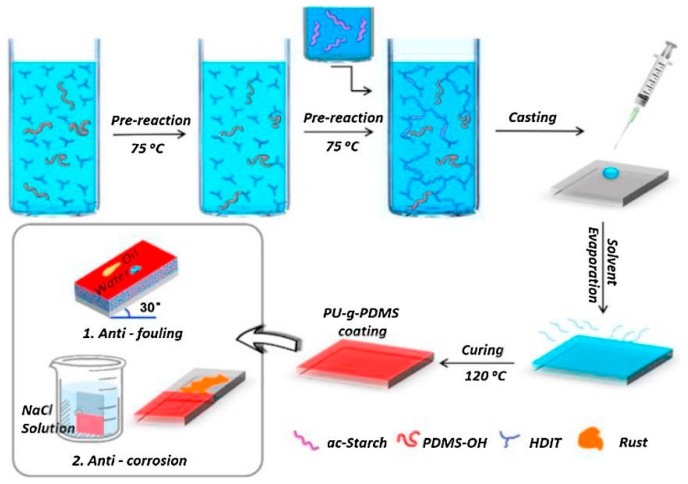
Preparation and characterization routes of PU-*g*-PDMS coatings [40].

**Figure 6 polymers-16-02155-f006:**
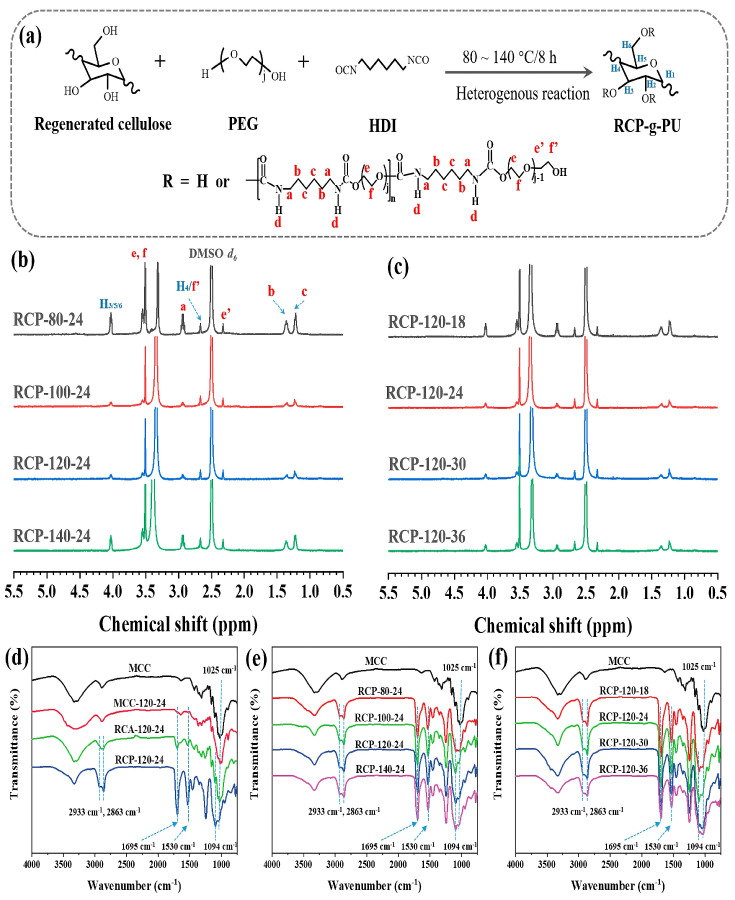
(**a**) Synthetic scheme of RCP−g−PU under heterogeneous conditions using amorphous RCP as a starting material; (**b**) 1H NMR spectra of RCP−g−PU synthesized at different reaction temperatures; (**c**) 1H NMR spectra of RCP−g−PU synthesized using different feeding ratios of PEG/AGU; and (**d**–**f**) ATR−IR spectra of MCCs and cellulose−g−PU prepared under various conditions [47].

**Figure 7 polymers-16-02155-f007:**
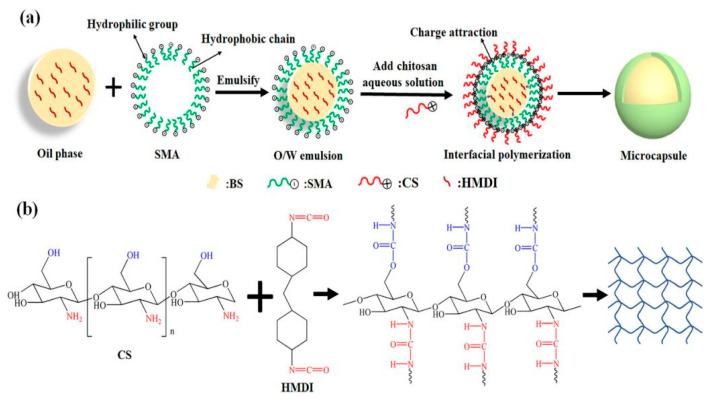
(**a**) Formation process of MicroPCMs prepared via interfacial polymerization and (**b**) chemical structures of CS and HMDI and the reaction formula of -NH_2_/-OH and -NCO [50].

**Figure 8 polymers-16-02155-f008:**
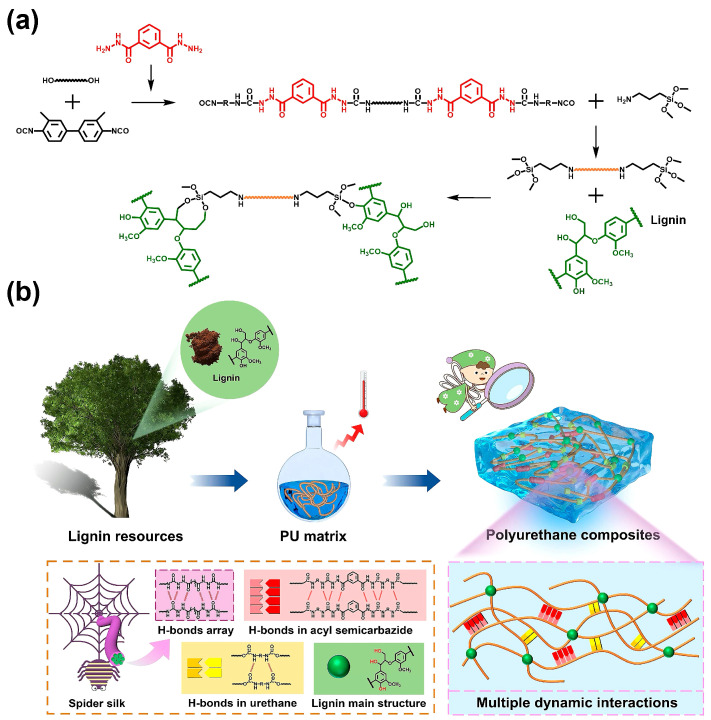
Synthetic route (**a**) and schematic diagram of the structural network (**b**) of LPUs [62].

**Figure 9 polymers-16-02155-f009:**
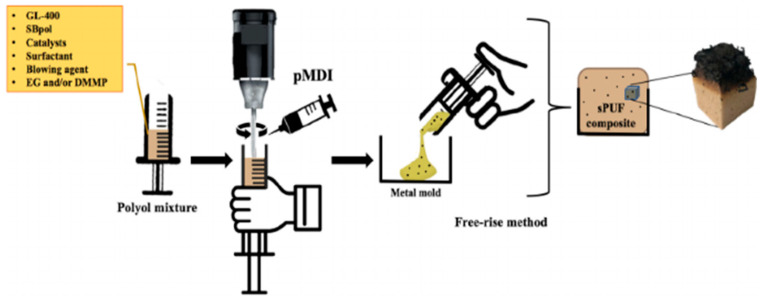
Schematic representation of the synthesis of sPUF composites [74].

**Figure 10 polymers-16-02155-f010:**
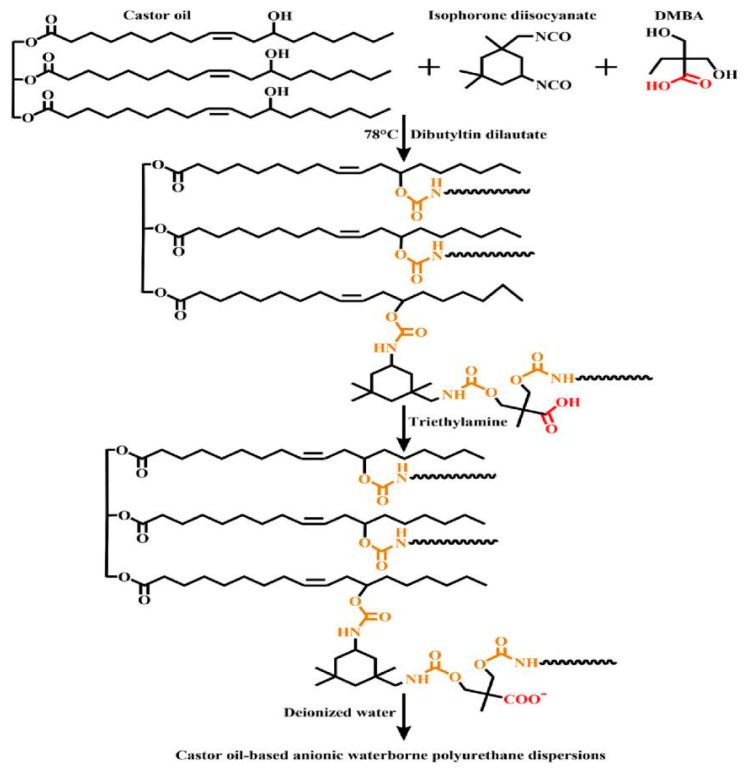
Synthesis route of castor oil-based anionic WPU dispersions [84].

**Figure 11 polymers-16-02155-f011:**
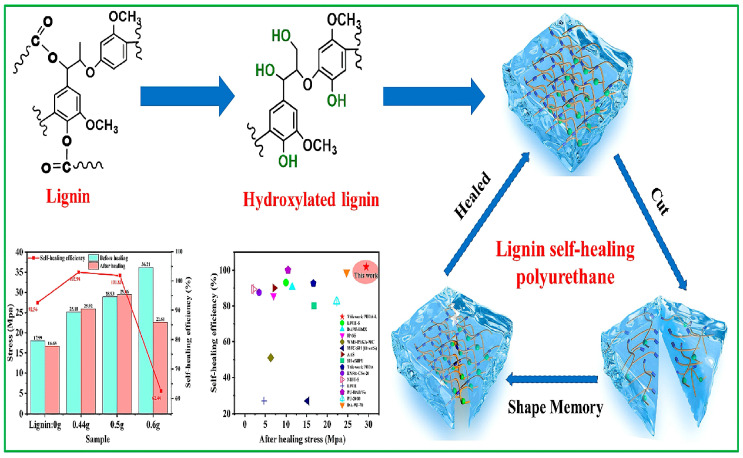
Preparation of self-healing polyurethane and its self-healing efficiency [93].

**Figure 12 polymers-16-02155-f012:**
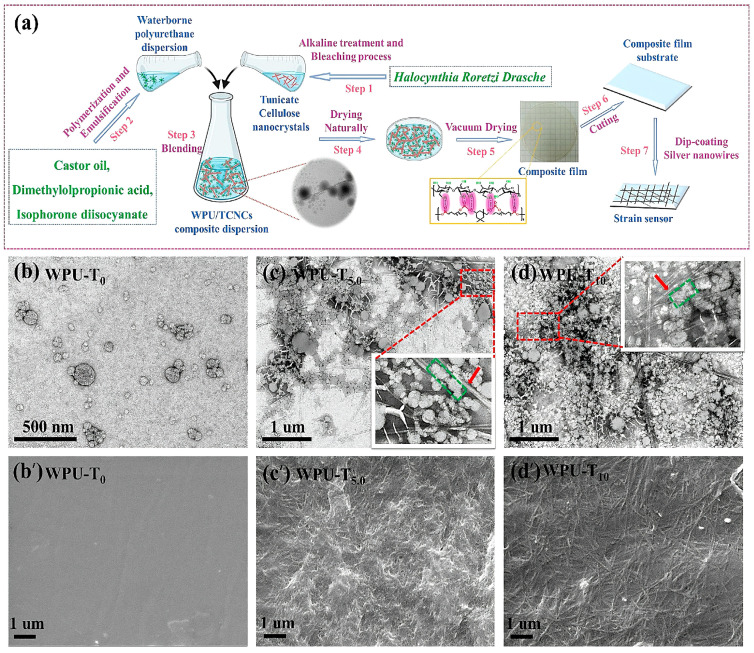
The process from raw material to strain sensor fabrication (**a**). TEM images of dispersions (**b**–**d**) and SEM images of films (**b’**–**d’**) [98].

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
