# Peer review of "Recent Advances in the Preparation and Application of Bio-Based Polyurethanes"

_polymers, 2024, doi:10.3390/polym16152155_

Round 1
Reviewer 1 Report
Comments and Suggestions for Authors
The article submitted for review is interesting and touches on important issues in the field of polyurethanes. There is a lot of new research going on in the field of polyurethanes, so the undertaking of writing a review article is important but also difficult.
In writing their review, the authors have chosen only a few topics in this field. I would like to see a clear statement at the beginning of the article as to which topics have been chosen and why.
The graphic abstracts copied from the cited articles are unfortunately unreadable. Also, please note that permission must be obtained from the authors to publish a copy of the figure.
Please standardise the spelling of bio-based or biobased in the article.
The article contains many spelling errors, missing spaces, improperly split words, etc., which make the article look carelessly prepared.
There is an error in figure 1 in the chemical structure - the double bond is between HO=C and the 'n' on the left side of the equation is missing.
Please be careful with statements ( line 57, 262) that biobased polyurethanes are perfectly biodegradable. This is not true. There are a whole host of polyurethanes that are derived from plant-derived materials, but when reacted with an isocyanate they are cross-linked polymers that do not degrade easily at all.
It is written (line 61) that isocyanates can be obtained from vegetable oils, but the article does not cite any items confirming this statement.
It is written (line 65) that castor oil has environmentally protective properties - please explain how ?
Author Response
1 In writing their review, the authors have chosen only a few topics in this field. I would like to see a clear statement at the beginning of the article as to which topics have been chosen and why.
Thank you for pointing this out.In the introduction we show why the two fields of polyurethane preparation and application were chosen. (line 52 to 56)
As follows:The research of bio-based polyurethane is currently in a stage of active development, and researchers are exploring various renewable biomass resources as raw materials for the preparation of bio-based polyurethane, optimizing the synthesis route and process conditions, so as to improve the preparation efficiency and quality stability of bio-based polyurethane. And explore its practical application value in various industrial and consumer applications.
2 The graphic abstracts copied from the cited articles are unfortunately unreadable. Also, please note that permission must be obtained from the authors to publish a copy of the figure.
Thank you for pointing this out.We have changed the definition of the image and have applied for permission from the copyright owner.
3 Please standardise the spelling of bio-based or biobased in the article.
Thank you for pointing this out.We have checked and corrected it.We unify it as bio-based
4 The article contains many spelling errors, missing spaces, improperly split words, etc., which make the article look carelessly prepared.
Thank you for pointing this out.We have revised and improved this
5 There is an error in figure 1 in the chemical structure - the double bond is between HO=C and the 'n' on the left side of the equation is missing.
Thank you for pointing this out.We have corrected the picture

6 Please be careful with statements ( line 57, 262) that biobased polyurethanes are perfectly biodegradable. This is not true. There are a whole host of polyurethanes that are derived from plant-derived materials, but when reacted with an isocyanate they are cross-linked polymers that do not degrade easily at all.
Thank you for pointing this out.This is caused by our overly one-sided description of it, which has been corrected in the article.(line 64, 295)
7 It is written (line 61) that isocyanates can be obtained from vegetable oils, but the article does not cite any items confirming this statement.
Thank you for pointing this out.We cited relevant literature to prove it.(line 72)
as shown in[18,19]
8 It is written (line 65) that castor oil has environmentally protective properties - please explain how ?
Thank you for pointing this out.This is a description error caused by our incorrect interpretation of the article, the original meaning refers to the preparation of degradable castor oil polyurethane through environmentally friendly methods, has been corrected, thank you for pointing out.
We tried our best to improve the manuscript and made some changes marked in the yellow highlighting in revised paper which will not influence the content and framework of the paper. We appreciate for your warm work earnestly, and hope the correction will meet with approval. Once again, thank you very much for your comments and suggestions.
Reviewer 2 Report
Comments and Suggestions for Authors
The article presented by the authors is interesting. It is an important contribution to the growing trend in the market for aromatic hydrocarbons used for chemical synthesis.
The paper lacked general information about aromatic hydrocarbons generated by chemical syntheses. Their use and importance in industry (production statistics). This information, in my opinion, should be in the introduction.
Please verify the form of citing figures in the text. In my opinion, the guidelines say Figure and not Fig.
Verses 459 to 465 should not be in the proposals. Proposals should include information that is aggregated such as the problems facing biomass polyurethanes. Progress, etc.
Author Response
1 The paper lacked general information about aromatic hydrocarbons generated by chemical syntheses. Their use and importance in industry (production statistics). This information, in my opinion, should be in the introduction.
Thank you for pointing this out. We have considered your suggestion deeply and consider it very instructive for our subsequent work. The main purpose of this article is to introduce the preparation and application of bio-based polyurethanes, which may be relatively less about the synthesis of aromatics. Since we do not have a deep understanding of the aromatics related materials, we are worried that we may provide misleading information in the article, so we choose not to add relevant information for the time being after fully considering your comments. Thank you very much for your valuable advice, and we will further explore related aspects in the future work. Thank you again and hope to get your forgiveness.
2 Please verify the form of citing figures in the text. In my opinion, the guidelines say Figure and not Fig.
Thank you for pointing this out.We correct Fig to Figure.
3 Verses 459 to 465 should not be in the proposals.Proposals should include information that is aggregated such as the problems facing biomass polyurethanes. Progress, etc.
Thank you for pointing this out.We have made changes to this aspect of the content.(line 541 to 551)
Such as: As a kind of sustainable and environmentally friendly high-performance material, bio-based polyurethanes have extensive research progress and application prospects, which provides an important choice and development direction for replacing traditional petroleum-based polyurethane. Many companies and research institutions are actively promoting the commercialization process of bio-based polyurethane, although there are still challenges in mass production and cost-effectiveness, but with technological advances and market demand growth, it is expected that more bio-based polyurethane products will be put into the market in the future. This article summarizes the research progress for plant oils, polysaccharides, and lignins in the preparation of biopolyurethane; introduces the current application statuses of bio-based polyurethanes across various fields; and explores the development and application prospects there of.
We tried our best to improve the manuscript and made some changes marked in the yellow highlighting in revised paper which will not influence the content and framework of the paper. We appreciate for your warm work earnestly, and hope the correction will meet with approval. Once again, thank you very much for your comments and suggestions.
Round 2
Reviewer 1 Report
Comments and Suggestions for Authors
In Figure 1, the letter n is still missing before the isocyanate. In this case it looks as if one molecule of isocyanate is reacting with n molecules of polyol.
Author Response
Reply
1 In Figure 1, the letter n is still missing before the isocyanate. In this case it looks as if one molecule of isocyanate is reacting with n molecules of polyol.
Thank you for pointing this out.This was an oversight on our part and we have corrected the picture. Thank you for pointing out our error and we apologize.(page 2)

We tried our best to improve the manuscript and made some changes marked in the yellow highlighting in revised paper which will not influence the content and framework of the paper. We appreciate for your warm work earnestly, and hope the correction will meet with approval. Once again, thank you very much for your comments and suggestions.